REGISTERED REPORT PROTOCOL

# Application and utility of a clinical framework for spinally referred neck-arm pain: A cross-sectional and longitudinal study protocol

**Camilla Kapitza**[1]*, **Kerstin Lüdtke**[2], **Brigitte Tampin**[1,3,4], **Nikolaus Ballenberger**[1]

**1** Faculty of Business, Management and Social Sciences, Department Movement and Rehabilitation Science, Hochschule Osnabrueck, University of Applied Sciences, Osnabrueck, Germany, **2** Department of Health Sciences, Academic Physiotherapy, Pain and Exercise Research Luebeck (P.E.R.L), Luebeck, Germany, **3** Department of Physiotherapy, Sir Charles Gairdner Hospital, Perth, Western Australia, Australia, **4** School of Physiotherapy and Exercises Sciences, Curtin University, Perth, Western Australia, Australia

These authors contributed equally to this work.
* c.kapitza@hs-osnabrueck.de

This is a Registered Report and may have an associated publication; please check the article page on the journal site for any related articles.

## Abstract

### Background

The clinical presentation of neck-arm pain is heterogeneous with varying underlying pain types (nociceptive/neuropathic/mixed) and pain mechanisms (peripheral/central sensitization). A mechanism-based clinical framework for spinally referred pain has been proposed, which classifies into (1) somatic pain, (2) neural mechanosensitivity, (3) radicular pain, (4) radiculopathy and mixed pain presentations. This study aims to (i) investigate the application of the clinical framework in patients with neck-arm pain, (ii) determine their somatosensory, clinical and psychosocial profile and (iii) observe their clinical course over time.

### Method

We describe a study protocol. Patients with unilateral neck-arm pain (n = 180) will undergo a clinical examination, after which they will be classified into subgroups according to the proposed clinical framework. Standardized quantitative sensory testing (QST) measurements will be taken in their main pain area and contralateral side. Participants will have to complete questionnaires to assess function (Neck Disability Index), psychosocial factors (Tampa Scale of Kinesiophobia, Pain Catastrophizing Scale, Depression, anxiety and stress scale), neuropathic pain (Douleur Neuropathique 4 Questions, Pain-DETECT Questionnaire) and central sensitization features (Central Sensitization Inventory). Follow-ups at three, six and 12 months include the baseline questionnaires. The differences of QST data and questionnaire outcomes between and within groups will be analyzed using (M)AN(C)OVA and/or regression models. Repeated measurement analysis of variance or a linear mixed model will be used to calculate the differences between three, six, and 12 months outcomes. Multiple regression models will be used to analyze potential predictors for the clinical course.

**Data Availability Statement:** All relevant data are in the paper and Supporting Information files. Data that will be collected until 2021 cannot be shared publicly because of its sensitivity (patient data), but will be made available on request, for e.g. meta-analyses, by Camilla Kapitza (c.kapitza@hs-osnabrueck.de) or the ethic committee from Hochschule Osnabrueck, University of Applied Sciences (Ethikkommission@hs-osnabrueck.de).

**Funding:** Camilla Kapitza is supported by the Hochschule Osnabrueck, University of Applied Sciences and she received a grant from Physio Deutschland, Deutscher Verband für Physiotherapie (ZVK) e.V.

**Competing interests:** The authors have declared that no competing interests exist.

## Conclusion

The rationale for this study is to assess the usability and utility of the proposed clinical framework as well as to identify possible differing somatosensory and psychosocial phenotypes between the subgroups. This could increase our knowledge of the underlying pain mechanisms. The longitudinal analysis may help to assess possible predictors for pain persistency.

## Introduction

Neck pain is a large health problem worldwide [1,2]. The point prevalence of neck pain ranges from 5.9% to 38.7%, the annual prevalence ranges from 16.7% to 75.1% [3]. Neck pain can radiate into the arm due to various underlying pain types and -mechanisms making it heterogeneous in clinical signs and symptoms [4–6]. We use the term pain type as a collective term for the three pain types (nociceptive, neuropathic and nociplactic pain) identified by the IASP according to Mitchell et al. [6] and the term pain mechanisms for the mechanisms which are underlying peripheral and central sensitization. On the one hand, patients may present with dominant nociceptive neck-arm pain caused by activation of the nociceptors in muscles, joints, ligaments, fascia, tendons and the connective tissues of a nerve. Activation of nociceptors in nerve connective tissues may cause clinical signs of heightened nerve mechanosensitivity [7–9]. On the other hand, patients may present with dominant neuropathic pain, defined as pain as a direct consequence of a lesion or disease affecting the somatosensory system [10,11]. Varying terminologies are used for spinally referred arm pain and there seems to be a lack of consensus on their definitions. For example, terms such as radiculopathy and radicular pain are often used synonymously in the literature, although they are different entities [7]. Radiculopathy is defined as sensory or motor deficit caused by a conduction block of a spinal nerve or its nerve root [7], whereby radicular pain is evoked by ectopic discharges radiating from a dorsal root or its ganglion. Often radicular pain occurs together with a radiculopathy but it can also stand alone or occur together with nociceptive pain [5,7,9]. The clinical profile of these different pain types is sometimes difficult to disentangle based on the localization and pain character. Additional measurements are required to define the underlying pain type [5].

Quantitative sensory testing (QST) assists in the interpretation of pain types and—mechanisms underlying clinical pain presentations [5,12–14]. It allows the assessment of function of all somatosensory modalities according to the different sensory nerve fibers (Aβ, Aδ and C fibers) and the documentation of a loss of function (hypoesthesia) or gain of function (hyperalgesia, allodynia) [13,15,16]. Based on QST, differences in somatosensory profiles between patients with C6/7 cervical radiculopathy and patients with C6/7 radicular pain without radiculopathy have been documented [5,17]. Patients with cervical radiculopathy were characterized mainly by a loss of function, consistent with nerve root damage and associated neuropathic pain, whereby patients with radicular pain were characterized by a gain of function and likely nociceptive pain [5,17]. The latter group also demonstrated clinical signs of heightened nerve mechanosensitivity, which suggests that the subgroup of 'heightened nerve mechanosensitivity' and 'radicular pain' can occur as a mixed pain presentation [9]. Moloney et al., reported mixed somatosensory presentations of sensory loss, hypersensitivity and no sensory abnormality in participants with cervical radiculopathy, but also in nonspecific arm pain [18]. These findings demonstrate the heterogeneity of neck-arm pain and highlight the need for further studies that examine the somatosensory profiles of different subgroups of neck-arm pain to understand the underlying mechanism and involved parameters.

Classifications can be helpful to define subgroups and the dominant or mixed pain type to guide the clinical decision making [19]. Studies have shown that tailored treatment is more effective than standard therapy in patients with low back pain [20–25]. Therefore, a classification should form subgroups that differ in the management required, as demonstrated by Schäfer et al. [26]. Schäfer et al. reported that patients with spinally related leg pain and heightened neural mechanosensitivity showed better results after neural mobilization treatment than the other subgroups [25]. Similarly, there is support for a mechanism-based management approach for some neuropathic pain conditions, i.e. better outcomes were achieved with subgrouped pharmacological treatments based on certain QST profiles [27].

Classification models for neck and neck-arm pain exist, but differ in their subgrouping criteria [20,28–32]. Some classifications focus on the stage of disorder and divide patients into acute, subacute and chronic [33–35], others classify patients in specific and nonspecific neck pain [32,33]. Some authors use criteria such as localization, duration of symptoms, episodes, pain severity and impairment [36] or different treatment-based models to classify patients [23,37–40]. However, these classification models do not address the source of symptoms and the underlying pain mechanisms, which both are important to consider in the management of patients with neck-arm pain. An evidence based clinical framework is required, which can be used in intervention studies to assess the effectiveness of treatment targeted at the underlying pain mechanisms for neck-arm pain. Schmid and Tampin proposed a mechanism-based clinical framework for spinally referred pain, taking into account pain types and—mechanisms as well as the clinical presentation [9]. The clinical framework distinguishes between, spinally referred leg pain without neurological deficits (somatic pain, heightened neural mechanosensitivity, radicular pain) and with neurological deficits (radiculopathy) as well as mixed pain presentations [9,41]. This clinical framework has been transferred to neck-arm pain [41], however, the application in patients with neck-arm pain has not yet been investigated. This clinical framework for spinally referred pain will be used to build the subgroups for investigating the somatosensory and psychosocial profiles of neck-arm pain patients in our study. The advantage of this framework over other mechanism-based approaches [20,28–31,42,43] is the precise terminology, the differentiation between radicular pain and radiculopathy and—based on this—a differentiated mixed pain presentation.

Predictors are important to assess the prognosis of pain conditions. Various clinical predictors of pain persistency in patients with neck and neck-arm pain have been reported [14,27,44–50]. These include psychological and cognitive-behavioural factors such as post-traumatic stress and pain catastrophizing in patients with whiplash and subacute neck pain [48], an initial high level of self-reported pain and disability [45,49], older age and a history of other musculoskeletal disorders in nonspecific neck-arm pain [45,49,51]. Poor muscle endurance as well as depressed mood were factors for the recurrence of pain in nonspecific neck pain [46]. QST parameters, such as cold hyperalgesia, was a significant predictor of poor outcome at long-term follow up in patients with whiplash [14] and dynamic QST (wind up ratio and cognitive pain modulation) in chronic pain conditions (fibromyalgia, nonspecific chronic back pain, chronic widespread pain) [27]. One single study, assessing patients with chronic neck and neck-arm pain with QST (CPT, PPT), clinical tests (neurodynamic tests), psychosocial factors (PCS, DASS-21), functional questionnaires (NDI) and neuropathic screening tools (SLANSS), demonstrated that baseline neck disability, comorbidities and higher psychological distress contributed to predicting disability at 12 months [44]. To date, this is the only study that has collected quantitative sensory and clinical tests to investigate potential predictors of chronic neck pain [44]. However, no study has included clinical measurements (e.g. active and passive cervical movement impairments, neurodynamic tests) as well as the somatosensory and psychosocial profile based on classified subgroups of different underlying pain disorders, to investigate potential predictors, as proposed in this study.

Hence, the overall goal of the current study is the application and evaluation of a clinical framework for spinally referred pain in patients with neck-arm pain, the assessment of their somatosensory and psychosocial profile as well as their clinical course over time.

Specifically, we aim:

(I) to investigate differences in somatosensory and psychosocial characteristics between subgroups that are based on a mechanism-based clinical framework;

(II) to compare side differences of somatosensory characteristics within subgroups;

(III) to track the course over time of pain, functional behavior and psychosocial parameters in each subgroup at three, six and 12 months;

(IV) to determine parameters at baseline that may predict clinical course over time.

## Method

### Ethics

The study has received ethical approval from the Ethics Committee of the University of Applied Sciences Osnabrück (HSOS/2019/2/2) and adheres to the ethical guidelines of the Declaration of Helsinki [52]. All patients will be asked to sign an informed consent form prior to participation.

### Designs

This is a prospective cohort study with cross-sectional and longitudinal analysis. Individuals with neck-arm pain will be recruited and divided into distinct subgroups according to the clinical framework for spinally referred pain (Fig 2) [9]. They will be compared with respect to somatosensory and psychosocial characteristics by a cross-sectional analysis (objective I and II). Subsequently, the same individuals will be followed over time in order to evaluate the clinical course and to identify potential predictors (objective III and IV). The test procedure is shown in Fig 1.

### Setting

Patients will be recruited from various physiotherapy and medical clinics as well as hospitals and radiology departments in and around Osnabrueck (Germany). Each subject fulfilling the eligibility criteria will be included. The measurements (clinical and QST testing) will take place at the INAP/O at the University of Applied Science Osnabrueck. The recruitment is anticipated to take place from July 2020 until probably December 2021, depending on the COVID pandemic. During the entire test procedure, the hygiene guidelines of the Robert Koch Institute and the University of Applied Sciences Osnabrueck will be followed.

### Participants

A screening interview via telephone will be conducted with each potential participant using a standardized questionnaire. This includes verifying the inclusion and exclusion criteria as well as providing information about the study. The inclusion criterion is unilateral spinally referred neck-arm pain in participants aged between 18–75 years. Exclusion criteria are previous spine surgery, current or previous systemic medical conditions (e.g. rheumatoid arthritis, diabetes, thyroid disease, HIV/AIDS, cancer), central nervous system disorder, complex regional pain syndrome, peripheral vascular disease, blood clotting disorder, pregnancy, psychiatric disease [15], presence of musculoskeletal shoulder, elbow or hand disorders in the last three months and an insufficient level of German or difficulty with communication which would prevent the participant to respond to QST measurements [5]. Following the screening, each suitable patient

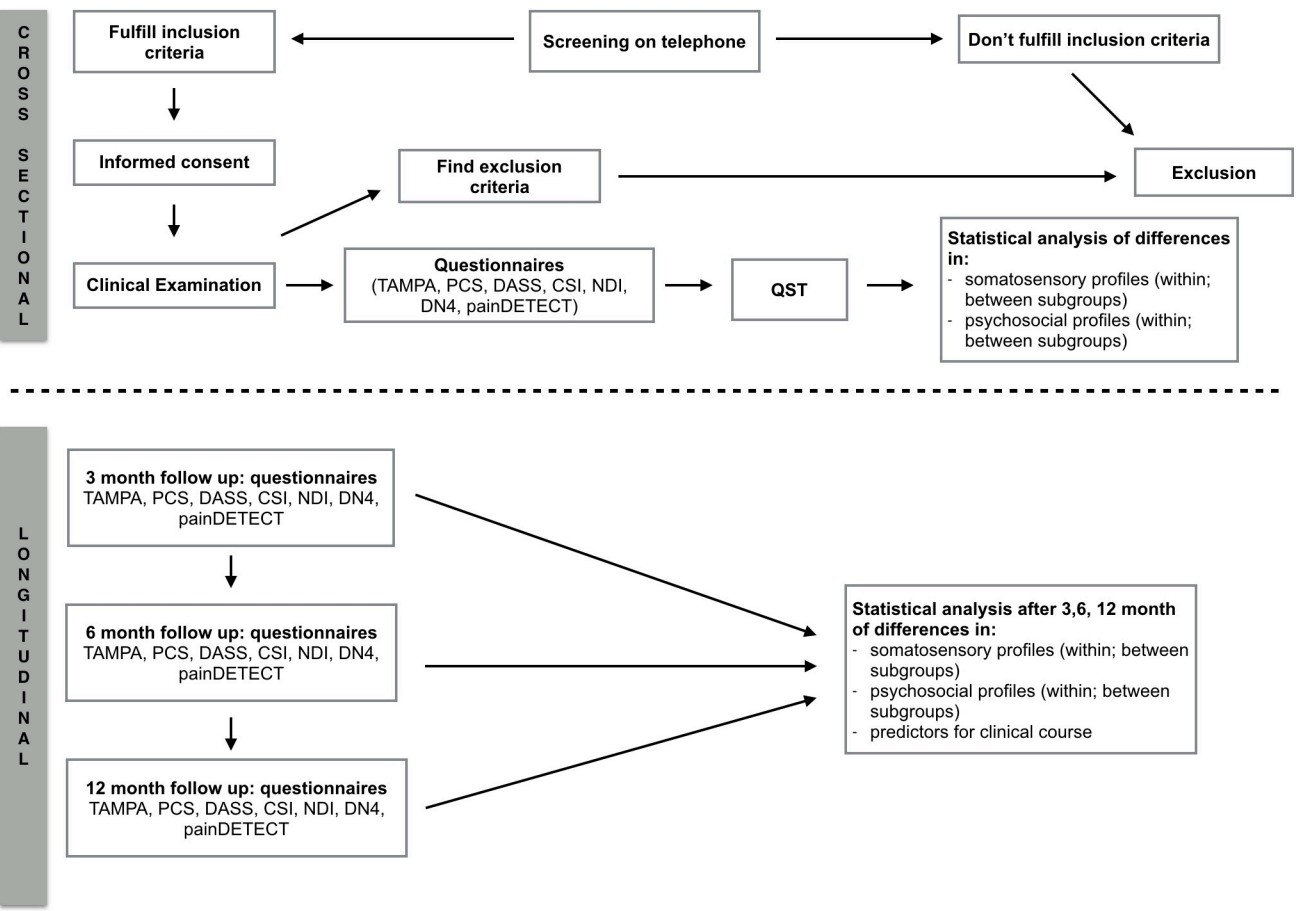

**Fig 1. Flow chart testing procedure.** NDI: Neck Disability Index, TSK: Tampa Scale of Kinesiophobia, PCS: Pain Catastrophizing Scale, CSI: Central Sensitization Inventory, DASS: Depression, anxiety and stress scale, DN4: Douleur Neuropathique 4 Questions, PD-Q: Pain*DETECT* Questionnaire, QST: Quantitative sensory testing.

will be given an appointment for the clinical examination as well the information sheet, the informed consent and a link to the study's homepage (www.nacken-armschmerzen.de).

## Testing protocol

Patients will undergo a clinical examination, complete questionnaires (see below) and undergo QST measurements. The complete examination (clinical and QST examination) will take 3 hours per subject. Participants are free to decide whether the complete examination takes place in one or in two appointments. Participants will need about 30 minutes to fill out the required questionnaires. There may be a maximum interval of seven days between the clinical and the QST examination in order to keep fluctuations in the pain and somatosensory presentation to a minimum. Should the clinical examination findings indicate the presence of any exclusion criteria, the participant will not proceed further in the study. Follow-ups will be conducted after three, six and 12 month.

## Clinical examination

The examiner (CK) will fill out a standardized clinical examination form (S1 File, Clinical examination form). This includes marking a body chart with the pain descriptors, pain

behavior and the history. In addition, co-morbidities, medications, special questions (red flags such as weight loss, spinal cord signs, vertebral artery) and sleeping behavior (numeric rating scale, 0 = good sleep; 10 = poor sleep) will be documented. The physical examination includes active (using a goniometer app [35,53,54]) and passive (examination of stiffness and pain [55]) movements of the cervical spine and shoulders. A thorough bedside neurological examination (BNE) will test the neurological integrity, including myotomal strength and reflex testing and sensory testing (soft touch, pinprick, warm/cold detection, cold hypersensitivity and vibration detection) [56–58]. Neural mechanosensitivity will be tested with the upper limb neurodynamic tests (ULNT) [57,59–61]. Based on the clinical examination findings, patients will be subgrouped according to the clinical framework for spinally referred pain (Fig 2) [9].

## Clinical framework for spinally referred pain

The clinical framework distinguishes between the presence of (i) somatic pain, (ii) neural mechanosensitivity, (iii) radicular pain and (iv) radiculopathy. Each of these presentations can occur in isolation but also in coexistence [9]. In the absence of neurological deficits/signs of a nerve lesion, the underlying pain type is, by definition, a nociceptive pain [7,9,10]. However,

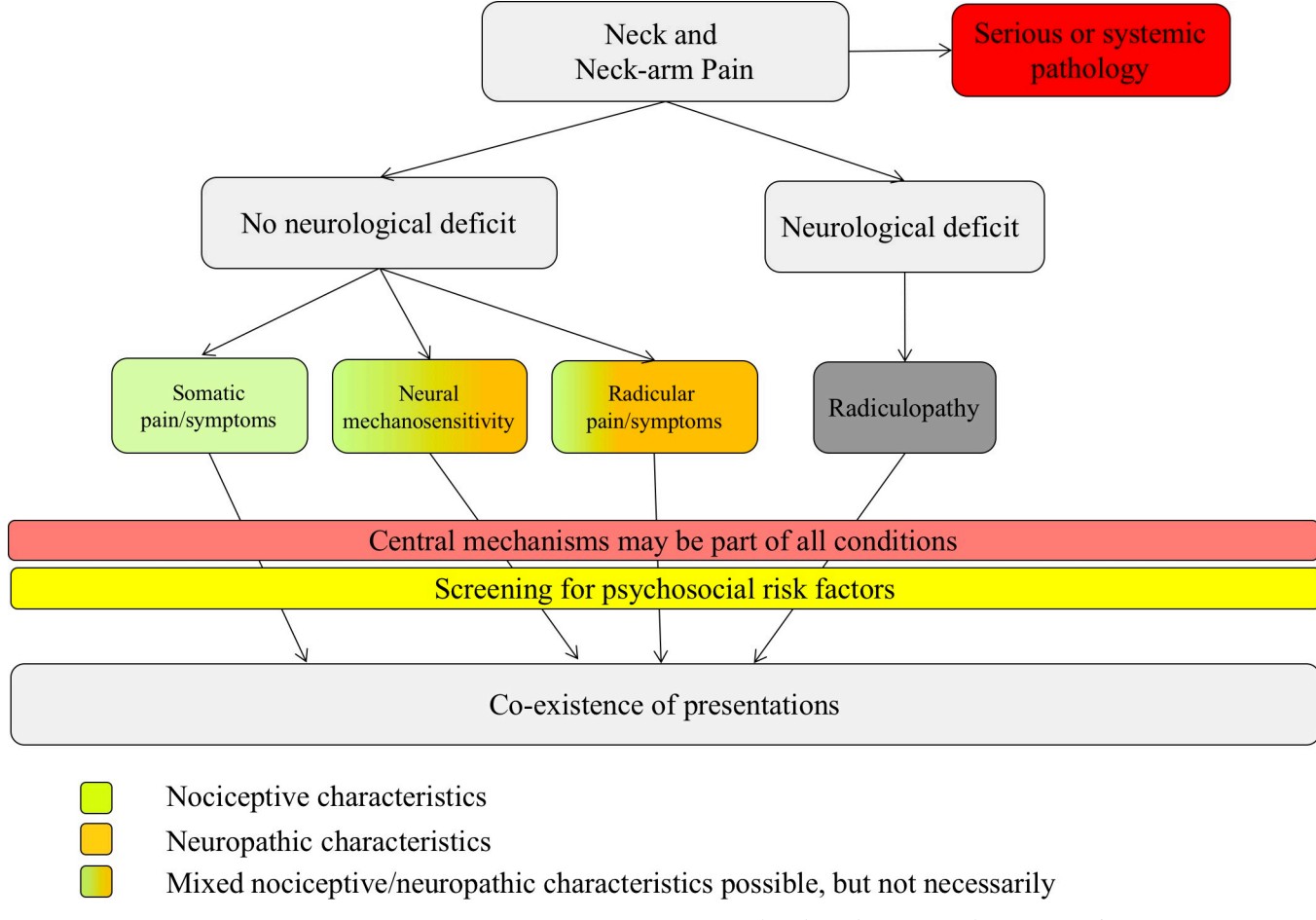

**Fig 2. Clinical framework of spinally referred neck-arm pain, adapted from reference.** The color code represents characteristics of nociceptive or neuropathic pain. The green color represents nociceptive pain characteristics, the orange color represents neuropathic pain characteristics (e.g. paraesthesia, numbness, pain descriptors such as burning pain, pain attacks). Green and orange represent mixed pain characteristics. The dark grey color represents a radiculopathy, which is not defined by pain, but by the presence of a neurological deficit.

some conditions present with neuropathic pain features, e.g. pain descriptors such as tingling, pins and needles, burning, pain attacks and numbness, hence the color coding to present mixed pain types.

Participants will be subgrouped into one of four classification groups (or mixed groups) based on their clinical examination outcomes:

- **Somatic referred pain**: neurological integrity test and tests for neural mechanosensitivity are normal.

- **Neural mechanosensitivity**: neurological integrity test are normal, but neurodynamic tests show heightened neural mechanosensitivity (ULNT's) [57].

- **Radicular pain**: the description of pain strongly suggests involvement of a nerve root (burning, pins and needles, shooting pain, electric shock, cold pain etc.); sensory tests may demonstrate hyperalgesia/allodynia. The neurological integrity tests show no pathological findings (loss of function); neurodynamic tests are normal.

- **Radiculopathy**: there is a myotomal or dermatomal neurological deficit present [57]. Since the 'radiculopathy group' is not defined by pain it has to be grouped with one of the subgroups [9].

- **Mixed pain:** there is a mixed pain presentation of the different pain types with or without neurological deficits and with or without heightened neural mechanosensitivity.

## Questionnaires

Patients will be asked to complete the following questionnaires before the QST appointment:

The Tampa Scale for Kinesiophobia (TSK) measures the fear of movement. The internal consistency showed an acceptable Cronbach's α of α = 0.73 and an good ICC (2.1) of 0.81 [62,63].

The depressions- anxiety- stress- scale (DASS) [64] examines 21 items, seven for each parameter. Reliability showed a good Cronbach's α of α = 0.88 for depression, acceptable α = 0.76 for anxiety and good α = 0.86 for stress. The sensitivity was 77% and the specificity 83%.

The Neck Disability Index (NDI) monitors intensity of pain, activities of daily living, headaches, concentration, work, driving a car, sleeping and leisure activities [65]. The validation of the German version resulted in a good Cronbach's α of 0.81, a correlation coefficient of r = 0.80 and a good ICC (2.1) of 0.81 [65].

The Pain Catastrophizing Scale (PCS) documents feelings and thoughts in painful scenarios in people with musculoskeletal pain [66]. Cronbach's α was excellent with an α = 0,92 and a good ICC (2.1) = 0,80 [66].

The Central Sensitization Inventory (CSI) is a screening instrument for central pain sensitization [67]. Reliability and validity were rated as high with r = 0.817 and a good Cronbach's α = 0.879 [68]. The cultural validation of the German version is currently in the operationalization phase. A pretest was successfully completed and made available to this study [69].

Douleur neuropathique 4 (DN4) is a screening questionnaire for neuropathic pain [70]. It includes sensory descriptors (pain quality and symptoms in the pain area) and a sensory examination (hyperesthesia for touch and pinprick, and for brush contact). Kappa values were between substantial 0.70 and almost perfect 0.96 [70].

The Pain*DETECT* (PD-Q) is a screening questionnaire for neuropathic pain [71]. The sensitivity, specificity and positive predictive accuracy, were 85%, 80% and 83%. It consists of 7 weighted sensory descriptors [71].

## Quantitative sensory testing (QST)

Standardized QST will be performed according to the reliable QST protocol of the German Research Network on Neuropathic Pain (DFNS) by a second examiner who will be blinded towards the subgroup classification process, the outcome of the questionnaires and the diagnosis [72]. The protocol includes seven tests that assess 13 different somatosensory parameters. Thermal thresholds will be measured with a MSA Thermal Stimulator (Somedic). The baseline temperature will be set at 32˚, cutoff temperatures are 5˚C and 50˚C. The following characteristics of temperature sensation will be recorded: cold (CDT) and warm detection threshold (WDT). The number of paradoxical heat sensations during the procedure of alternating warm and cold stimuli (TSL) and cold (CPT) and heat pain threshold (WPT) will be measured. The mean threshold temperature from 3 measurements will be calculated.

The mechanical detection threshold (MDT) will be tested with a standardized set of modified von Frey hairs (forces between 0,25 and 512 mN). The mechanical pain threshold (MPT) will be tested with a set of 7 weighted pinprick stimulators (8 to 512 mN). For measurements of MDT and MPT, five ascending and descending series will be applied and the geometric mean will be calculated.

The stimulus response function tests how painful the patient rates various pinprick stimuli, using the same weighted pinprick stimulators as for MPT: mechanical pain sensitivity (MPS) and light touch stimuli, using cotton wool tip, cotton wisp and a brush: dynamic mechanical allodynia (ALL). Subjects will be asked to give a pain rating for each stimulus on a NRS (0 = no pain, 100 = most intense pain imaginable). The wind-up ratio (WUR) to repetitive pinprick stimuli will be tested. The patient will assess the pain intensity of a single needle stimulus on a scale of 0–100 and compare it with a series of 10 consecutive needle stimuli applied at a 1/s rate. The vibration detection threshold (VDT) tests the ability to perceive vibration stimuli and will be tested with a standardized Rydel-Seiffer vibration fork (64 Hz, 8/8 scale). Pressure pain threshold (PPT) tests the pain intensity of blunt pressure. The patients will be asked to push a button when the sensation changed from one of pressure to one of pressure and pain. PPT will be performed using an algometer (Somedic). Measurements will be taken from the maximal pain area nominated by the patient and the contralateral side.

Testing of the full QST protocol will take approximately 30 minutes per test area.

Healthy control reference data for the neck-arm areas will be obtained in another parallel study conducted at the University of Applied Sciences Osnabrueck. Age-, and gender matched QST norm data will be collected in healthy participants for each body region tested in patients. For each body area, reference data of 16 subjects (8 female, 8 male) per age decade will be collected [72]. The reference data will be used to calculate z-scores.

## Follow-up questionnaires

The three, six and 12 months follow-up will be conducted via postal hard copy and lime survey (Version: 3.22.210 + 200804).

It will include the following questionnaires:

- Tampa Scale for Kinesiophobia (TSK) [62,63,73]

- Depressions- anxiety- stress- scale (DASS) [64]

- Neck Disability Index (NDI) [65,74]

- Pain Catastrophizing Scale (PCS) [75]

- Central Sensitization Inventory (CSI) [68]

- Douleur Neuropathique 4 (DN-4) and Pain*DETECT*: Screening tools for neuropathic pain [70,71].

- A questionnaire capturing pain behavior, disability and interventions (operations, physiotherapy, exercises, pain management, infusions etc.) carried out.

- Numeric rating scales for current pain, maximal and minimal pain.

- Patient Global Impression of Change scale [51].

## Sample size

Calculation of the sample size was processed with G-Power (Version: G*Power 3.1.9.4.). A sample size of 45 per group is required to detect a medium sized effect based on the calculation of an analysis of variance with a significance level of 0.05 and a power of 80% The assumption of medium sized effects derives from the study by Ottiger-Boettger et al. [76], in which small to large-effect-sized group differences were detected with respect to QST measurements between patient groups with non-specific neck-arm pain.

## Statistical analysis

All data analysis will be performed with SPSS (Version: 26) and R (Version: 3.6.3.). Characteristics of study population depending on subgroups will be explored with descriptive statistics, ANOVA, Kruskal-Wallis-Test or $Chi^2$ Test depending on the scaling of variables and statistical assumptions of parametric testing. Objective (I,II) will be addressed with a cross-sectional analysis. For calculation of differences in somatosensory and psychosocial characteristics between and within subgroups, (M)AN(C)OVA and/or (hierarchical) regression models will be used. The choice of the final model depends on, among others, the presence of potential confounders and/or the need to take into account more than one measurement from the same subject (e.g. in case of comparing left and right side within the same subject). For the latter hierarchical regression models are appropriate. In case of violation of statistical assumptions data transformation (e.g. log-transformation) or non-parametric testing will be considered such as Kruskal-Wallis-Test, Friedman Test and non-parametric or robust regression. QST data will be log-transformed prior to statistical analysis, except those data which are normally distributed as raw data [72]. QST data will be z-transformed using the following calculation: z-score = (mean single patient—mean healthy controls/SD healthy controls) [72]. Potential Confounder variables (e.g. age, gender) will be considered in the model.

Course over time (Objective III) and its predictors (Objective IV) will be addressed by longitudinal analysis. The within and between groups difference at the four time points will be determined. MANOVA, repeated-measures ANOVA, and mixed regression models will be used to model course over time, differences of subgroups and their interaction (subgroup membership x course over time). The independent variables are psychosocial questionnaires (TAMPA, DASS, PCS), QST data, neuropathic screening questionnaires, central sensitization features (CSI), pain intensity and the classified subgroup at baseline. The dependent variables are presence and number of pain episodes, duration of symptoms and the NDI at respective time points. However, due to the lacking of comparable studies our planed experiment is partly of exploratory nature. As a consequence the exact plan of analysis is difficult to specify in beforehand. Yet, the a priori planned structure of the experiment will not be subject to change (number of measurements, measurements time points, assessments, classification criteria, etc.).

The calculated sample size is expected to be achieved by end of 2021. Follow-up data will then be collected according to the defined point of measurement. After the primary statistical analysis, the data will be considered again in the secondary analysis by a cluster analysis.

## Limitations of the study

A limitation of this study is that the patient allocation to subgroups is based on clinical examination findings. Additional instrumental measurements (e.g. MRI, nerve conduction studies, Somatosensory Evoked Potentials) to validate clinical findings will unlikely be available. Furthermore, the clinical examination and patient classification will be performed by one examiner. An assessment by a second examiner would enhance the validity of the study, however there are resource limitations, plus repeated assessment would impose a considerable burden to the patient.

## Conclusion

This study will evaluate a newly proposed mechanism-based clinical framework for neck-arm pain which classifies patients according to their clinical presentations as well as underlying pain types and pain mechanisms. Comprehensive sensory and clinical profiling will assist in the characterization of each subgroup and the possible predictive role of measured parameters for pain and disability persistency over 12 months will be investigated.

## Supporting information

**S1 File. Clinical examination form.**
(DOCX)

## Acknowledgments

We thank Christina Krone for her assistance in QST assessments and Sabrina Friehe for her assistance in follow-up assessments. We thank staff at INAP/O and Elbestrasse for their invaluable support. We thank Ellen Loock and Svenja Hardt for collecting reference QST data and all participants for supporting our study with their participation.

## Author Contributions

**Conceptualization:** Camilla Kapitza, Kerstin Lüdtke, Brigitte Tampin, Nikolaus Ballenberger.

**Data curation:** Camilla Kapitza.

**Formal analysis:** Camilla Kapitza, Kerstin Lüdtke, Brigitte Tampin, Nikolaus Ballenberger.

**Investigation:** Camilla Kapitza.

**Methodology:** Camilla Kapitza, Kerstin Lüdtke, Brigitte Tampin, Nikolaus Ballenberger.

**Project administration:** Camilla Kapitza.

**Resources:** Camilla Kapitza, Brigitte Tampin.

**Supervision:** Kerstin Lüdtke, Brigitte Tampin, Nikolaus Ballenberger.

**Validation:** Camilla Kapitza.

**Writing – original draft:** Camilla Kapitza.

**Writing – review & editing:** Camilla Kapitza, Kerstin Lüdtke, Brigitte Tampin, Nikolaus Ballenberger.

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
