## [Decision Letter · Decision Letter 0]

1 Sep 2020

PONE-D-20-20814

Evaluation of a mechanism-based classification for neck-arm pain: A cross sectional and longitudinal study

PLOS ONE

Dear Dr. Kapitza,

Thank you for submitting your manuscript to PLOS ONE. After careful consideration, we feel that it has merit but does not fully meet PLOS ONE’s publication criteria as it currently stands. Therefore, we invite you to submit a revised version of the manuscript that addresses the points raised during the review process.

The manuscript presents a **Registered Report Protocol: **an article describing the study design, rationale, timeline, proposed methodology for data collection and analysis, and where applicable ethical approval for the work. Registered Report Protocols report the study proposal prior to conducting experiments, data collection or patient recruitment, and they undergo peer review to ensure that the planned research will meet *PLOS ONE’s* publication criteria. Accepted Registered Report Protocols are published in the journal, and receive an in-principle accept for the future article reporting the results of the study after completion.

We look forward to receiving your revised manuscript.

Kind regards,

Alison Rushton

Academic Editor

PLOS ONE

Additional Editor Comments:

Please address the reviewers feedback below and ensure the protocol nature of the study is clear.

3. Please amend your list of authors on the manuscript to ensure that each author is correctly linked to an affiliation. Authors’ affiliations should reflect the institution where the work was done (if authors moved subsequently, you can also list the new affiliation stating “current affiliation:….” as necessary).

Reviewers' comments:

Reviewer's Responses to Questions

**Comments to the Author**

1. Does the manuscript provide a valid rationale for the proposed study, with clearly identified and justified research questions?

Reviewer #1: No

Reviewer #2: Yes

Reviewer #3: No

Reviewer #4: Yes

2. Is the protocol technically sound and planned in a manner that will lead to a meaningful outcome and allow testing the stated hypotheses?

Reviewer #1: No

Reviewer #2: Yes

Reviewer #3: No

Reviewer #4: Yes

3. Is the methodology feasible and described in sufficient detail to allow the work to be replicable?

Reviewer #1: Yes

Reviewer #2: Yes

Reviewer #3: No

Reviewer #4: Yes

4. Have the authors described where all data underlying the findings will be made available when the study is complete?

Reviewer #1: No

Reviewer #2: Yes

Reviewer #3: No

Reviewer #4: Yes

5. Is the manuscript presented in an intelligible fashion and written in standard English?

Reviewer #1: No

Reviewer #2: Yes

Reviewer #3: Yes

Reviewer #4: Yes

6. Review Comments to the Author

You may also provide optional suggestions and comments to authors that they might find helpful in planning their study.

Reviewer #1: 

The manuscript describes more of what will be done in a yet to be conducted study whereas the title gives the impression that the study had been done.

Authors need to pay attention to all details in the manuscript starting from the authors' list on page 1. Who among the authors is affiliated with Australia as indicated on line 17?

Why #a and #b instead of continuous numbering?

The abstract section as well as the introduction and method sections are each too long. This makes comprehension of each section difficult.

The method section is written in future tense instead of past tense.

Page 9 lines 222-224: Why is age not an inclusion criterion since pain perception could vary as a function of age?

Page 14: It is not clear from which reference study the sample size was determined.

Page 15 lines 366-373, if there could be need for non-parametric statistic as earlier indicated on the same page, why is this section referring only to parametric statistics?

How can the authors be sure that the study is feasible to come to the conclusion of what knowledge the study will contribute or the impact it will make?

Reviewer #2: 

Abbriviations are commonly not used in abstracts.

Reference no 1: Vos, T., et al.,m 2012 is based on data from 1990-2010. Maybe newer references are available? Same thing with reference no 3: Cote, P., et al., 2004 (also here newer date might be available.

page 7: the aim: Hence, the overall goal of the current study is the evaluation of a mechanism-based

classification for patients with neck-arm pain.

This study has not completed the sampling, thus not being able to evaluate this classification. This study has more characteristic of a protocol. The editor should determine if the aim should be modified or if protocols without results are allowed. Furthermore, a modification of the titel could be relevant, since an evaluation has not yet taken place in this study.

Page 9:Sampling and Inclusion criteria: unilateral pain is only inclusion criteria, but what about how long time the pain has been present (acute, subacute, chronic)? This is vital information in order to replicate and understand why and how this classification will be developed.

What about interventions that the included people may take up during the 12 months (fx physiotherapy, pain management, acupuncture etc.), how are these integrated/taken into account in the data?.

page 14: sample size: A sample size of 45 per group is 351 required to detect a medium sized effect......

Effect is not an aim of this study, maybe the authors could clarify why 45 subjects should attend each subgroup when intervention is not a part of this study, but rather the charateristics/symptoms of the patients/between patients.

Figure 2: A bit confusing that one box/characteristic is half green/half orange, and another one third green and two-third orange (they are actually both mixed). Can this be explained below the figure?

And the grey box should also be explained like the others colors below the figure (the pain characteristics).

Reviewer #3: 

It is an interesting topic but it would be great to add more rationale for doing this study: 1) how it would change rehabilitation/physiotherapy practices?; 2) What limitations are there for this study?

Reviewer #4: 

Review of Study Protocol: Evaluation of a mechanism-based classification for neck- arm pain. A cross sectional and longitudinal study

This is a well-conceived study protocol to evaluate a proposed classification system for neck-arm pain. I enjoyed reading it.

1. Does the manuscript provide a valid rationale for the proposed study, with clearly identified and justified research questions?

The paper provides a valid, and thorough rationale for the proposed study. The research questions are well formed and follow the logic of the background and introduction section.

In the abstract, the rationale is described by the single-sentence – “The rationale for this study is to assess the usability of the proposed classification system” - I would like to see this expanded to say that in addition to its usability the rationale is to assess its utility and robustness too by assessing the classification longitudinally in participants whose pain syndrome is expected to change over time,

In the introduction L112-113 I would like to have read why “The use of a mechanism-based classification should be applied to direct assessment and management [12].” In contrast to the other classification systems also described – could they not equally direct assessment and management?

The Schmid and Tampin classification approach is introduced as the preferred classification model you are deciding to utilise and evaluate which is logical as you make the argument that a mechanism-based approach is missing – but is the Schmid and Tampin approach the only one – are there others? And if so, why this one?

Paragraph (starting on line 123) introducing QST – I think you could be much more explicit here in arguing that including this approach will allow some resolution between mixed types of pain experience. If this is correct, I’d say make this assertion as a gaol of the study nearer the beginning of the paragraph – then go on to explain why. Similarly, in the paragraph preceding this one – the aim is mentioned at the end of the paragraph, and I think the manuscript would benefit from these statements being introduced earlier and therefore in a bolder way.

Overall the rationale is good, but I think it could be crafted to tell an even more compelling story with some minor edits and removal of passive voice – I have alluded to specific edits/typos below.

2. Is the protocol technically sound and planned in a manner that will lead to a meaningful outcome and allow testing the stated hypotheses?

The ethics and design sections are well written and clear. I think in Figure 1 the vertical title bar “Longitudinal Sectional” should just read “Longitudinal”. The protocol is technically sound, and the measurement and analytical approaches are justified and appropriate.

The sample size calculation sub-section is welcome (L349)– however it lacks detail in that it does not stipulate the target effect size the calculation is based on (the parameter, the type, or the magnitude), nor how the effect size was was determined. Reference to G-power should include detail of the software and the version number used for completeness.

A minor style point, but I think the software used should be detailed at the end of the paragraph, not as the first thing (likewise in the “statistical analysis” sub-section).

3. Is the methodology feasible and described in sufficient detail to allow the work to be replicable?

The methods are feasible. I note that sampling is due to have started in June 2020, and also note that no allusion to the possible effects prolonged social restrictions due to the COVID pandemic might have on recruitment, retention and practicality of measuring constructs that cannot be done with participants remotely – i.e. QST assessment - in Germany. If there has been any mitigation for these unforeseen circumstances, then I think they should be added to the protocol.

I note there is no allusion in the inclusion criteria for age; presumably the study is focussed on adults and therefore needs at least a minimum age. The exclusion criteria are comprehensive but appear to be an exclusion based on any count of pre-determined systemic pathologies (in contrast to say a comorbidity index threshold, or health-care utilisation threshold); I am not familiar with the specific pain syndrome under investigation in this paper so I do not know if there is room to justify this approach or the selection of the pathologies with due referencing, this would also defend the approach from the point of view of over-excluding and therefore only recruiting and testing a niche sub-set of participants– so my question would be are these exclusions typical in the field?. One particular query I have is “… elbow or hand disorders in the last months, …” – is this referring to musculoskeletal disorders or other disorders (neurological, or peripheral vascular for example), and a definitive number of months would make the exclusion clearer.

I think L216-218 “… as well as the information sheet, the informed consent and a link to the study's homepage (www.nacken-218 armschmerzen.de).” might be better as “… as well as familiarisation information including a written information sheet, the informed consent material and a link to the study's homepage …”

Informed consent information is clearly stated in the Ethics sub-section, so the repeated allusion to it in L219-20 feels unnecessary.

Testing Protocol subsection – the first sentence is lacking a point - when or where will this take place, will the listed tasks be undertaken in one appointment for example? The next sentences provide more detail, but I think this sub-section could be clearer.

L227 – suggest signposting the reader here that clinical self-reported questionnaire details are provided below

Clinical Examination sub-section – I would like to see more detail referred to either in the text or as a footnote (if the journal allows it) or as supplementary material. Specifically; what are the red-flag questions? what are the anchors of the sleeping numerical rating scale? The passive and active c-spine/shoulder complex exam – while these are referenced, I would like to see in the text what movements/planes are planned to be measured

The Classification system sub-section is welcome – a minor point is that while neural mechanosensitivity and neurodynamic tests are mentioned in the text prior to the bullet-points in this sub-section, the neurological integrity test is not and might confuse readers unfamiliar with the classification system – it might help to weave this into the narrative prior to the bullets at least in the clinical examination sub-section.

L296 “Kappa values were between 0.70 and 0.96 [70]” would be a stronger statement with a justification and an interpretation of these data

Statistical Analysis sub-section - L360-361 – “In case of violation of statistical assumptions data transformation or non-parametric testing will be considered” please provide details of what transformations will be considered or soften with “appropriate” transformations. L374-376 – this sentence is welcome but needs to be re-crafted, so it is clearer.

The Conclusion I think is excellent.

4. Have the authors described where all data underlying the findings will be made available when the study is complete?

Yes

5. Is the manuscript presented in an intelligible fashion and written in standard English?

Yes. I did find some trivial edits which are outlined below:

Abstract, L61 – is there a need to pluralise regression model to models here?

Introduction, L97 - “Studies showed that tailored treatment was more effective …” might be better as “Studies have shown that tailored treatment is more effective …”

Introduction, L103 – “condition” should be pluralised I think

Introduction, L128 –suggest change “were” to “have been”

Introduction L140-145 – while understandable, this is a long sentence to parse – it might help a reader to break into >1 sentence

Introduction L145 – I think there is a missing “of” before “… a new episode …”

Introduction L145 – I think “… and …” could be changed to “… as was …” to help this sentence.

Introduction L149 – I think the sentence starting “One single …” could be made clearer and link to the next sentence better; in fact I think “To date, there is only this one study that …” could be bolder and say “To date, this is the only study that …” in the next sentence

Introduction L155 – This final sentence could also be made bolder, perhaps choose another way of saying “to date” in it, and not end with a passive goal of the proposed study.

Introduction L161 – I would consider changing the rather passive opening to this paragraph (“It could be summarized that …”) with something more assertive.

Methods L200 – I am not familiar with “executive sample”, is this correct? Should it be a sample of convenience, or a volunteer sample

Methods L208 – I think “criterium” should be “criterion”

Methods L216 – tense consistency; I think “… is …” should be “… will be …”

Methods L220 – I think “..., in another parallel …” could be simply

Methods L268 – “Since the ‚radiculopathy group’ …” should be “Since the ‘radiculopathy group’ …”

Methods L277 (& L280; L287) – suggest to remove the second , the use of the word “good” for the value of 0.73 needs would be strengthened with a published precedent , I would also like to see the type of ICC referred to in this line for completeness (also for ICC on L285)

Methods L279 – “The Depressions- Anxiety- Stress- Scale (DASS) …”, I think this should read “The depression anxiety and stress scale (DASS) …” And, “… examines 21 Items …” should be “… examines 21 items …”

Methods L282 – I do not think the word “the” is needed in “… monitors intensity of the pain …”

Methods L293 – “The pretest was …” could be better stated as “A pretest was …”

Methods L304-305 the statement “… by a second blinded examiner toward the classified subgroup, …” would be clearer as “… by a second examiner who will be blinded to the subgroup classification process , …” or similar

Methods L306 - I think “… seven tests which tests 13 different …” should be “… seven tests that assess 13 different …”

Methods L307 – I think “… Baseline temperature is at …” could be better as “… Baseline temperature will be set at …”

Methods L312 – I think “… temperature of 3 measurements will be calculated.” could be “… temperature from 3 measurements will be calculated.”

Methods L321 – “Subjects are asked …” should be “Subjects will be asked …”

Methods L355-358 – This sentence does not make complete sense to me

Methods L358 – the word “answered” needs to be changed

Methods L360-361 – “In case of violation of statistical assumptions data transformation or non-parametric testing will be considered” please provide details of what transformations will be considered, or soften with “appropriate” transformations

Methods L368 – “… ANOVA of repeated measurements …” should be “… repeated-measures ANOVA …” I think.

6. Review Comments to the Author

You may also provide optional suggestions and comments to authors that they might find helpful in planning their study.

This is a valid protocol for a needed study and utilises both a cross sectional and longitudinal design. The protocol is sound, and my comments really are of style and some content detail which is missing in my opinion. With some minor revision, the manuscript protocol would be a welcome addition to the literature.

7. PLOS authors have the option to publish the peer review history of their article (what does this mean?). If published, this will include your full peer review and any attached files.

Reviewer #1: No

Reviewer #2: No

Reviewer #3: No

Reviewer #4: No

---

## [Author Response · Author response to Decision Letter 0]

20 Oct 2020

Reviewer Comment Response:

Reviewer 1 

The manuscript describes more of what will be done in a yet to be conducted study whereas the title gives the impression that the study had been done. 

Thank you for your comment.

We changed the title to make it more clear that the paper is a study protocol. We changed the title from: 

Study Protocol: Evaluation of a mechanism-based classification for neck-arm pain – A cross sectional and longitudinal study

 into: 

Application and utility of a clinical framework for spinally referred neck-arm pain : A study protocol of a cross-sectional and longitudinal study

Authors need to pay attention to all details in the manuscript starting from the authors' list on page 1. Who among the authors is affiliated with Australia as indicated on line 17?

Why #a and #b instead of continuous numbering? 

Thank you very much. We adjusted the title page according to the PLOS ONE guidelines. See in “revised manuscript with track changes”. Page 1 L1-21.

The abstract section as well as the introduction and method sections are each too long. This makes comprehension of each section difficult. 

Thank you! The abstract has been shortened to 300 words. We have revised the introduction and method sections. However, some reviewers asked to add further information, which made it difficult to shorten some sections. According to PLOS ONE author guidelines, manuscripts can be at any lengths; there are no restrictions on word counts.

The method section is written in future tense instead of past tense.

The method section is written in future tense, because we have planned the investigation this way and haven’t completed the study yet. We hope this has become clearer throughout the manuscript.

Page 9 lines 222-224: Why is age not an inclusion criterion since pain perception could vary as a function of age? 

We apologize for the oversight. The included age range is from 18-75 We decided on this relative large age range in order to capture a realistic patient sample.

The new sentence is:

Page 9/10 L226-227.

The inclusion criterium is unilateral neck-arm pain in participants aged between 18 – 75 years.

Page 14: It is not clear from which reference study the sample size was determined. 

Thank you for this comment. We refer to the study by Ottiger-Boettger et al, titled: “Somatosensory profiles in patients with non-specific neck-arm pain with and without positive neurodynamic tests“. The study is currently in press [1].

We calculated the sample size with G*Power 3.1.9.4., based on the selected statistics, we assume that we can operate with a medium sized effect. In the study by Ottiger-Boettger et al., small to large-effect-sized group differences were detected with respect to QST measurements. We added this information in the manuscript [1].

Adapted in the manuscript page 16, L 379-384:

Calculation of the sample size was processed with G-Power (Version: G*Power 3.1.9.4.). A sample size of 45 per group is required to detect a medium sized effect based on the calculation of an analysis of variance with a significance level of 0.05 and a power of 80%. The assumption of medium sized effects derives from the study by Ottiger-Boettger et al. [1], in which small to large-effect-sized group differences were detected with respect to QST measurements between patient groups with non-specific neck-arm pain.

Page 15 lines 366-373, if there could be need for non-parametric statistic as earlier indicated on the same page, why is this section referring only to parametric statistics?

Thank you, we added the missing non-parametric test. Page 16, L387-402.

The section reads now:

Characteristics of study population depending on subgroups will be explored with descriptive statistics, ANOVA, Kruskal-Wallis-Test or Chi²Test depending on the scaling of variables and statistical assumptions of parametric testing. Objective (I,II) will be addressed with a cross-sectional analysis. For calculation of differences in somatosensory and psychosocial characteristics between and within subgroups, (M)AN(C)OVA and/or (hierarchical) regression models will be used. The choice of the final model depends on, among others, the presence of potential confounders and/or the need to take into account more than one measurement from the same subject (e.g. in case of comparing left and right side within the same subject). For the latter hierarchical regression models are appropriate. In case of violation of statistical assumptions data transformation (e.g. log-Transformation) or non-parametric testing will be considered such as Kruskal-Wallis-Test, Friedman Test and non-parametric or robust regression. QST data will be log-transformed prior to statistical analysis, except those data which are normally distributed as raw data [2]. QST data will be z-transformed using the following calculation: z-score= (mean single patient - mean healthy controls/ SD healthy controls) [2]. Potential Confounder variables (e.g. age, gender) will be considered in the model.

How can the authors be sure that the study is feasible to come to the conclusion of what knowledge the study will contribute or the impact it will make? 

Thank you. We can’t be sure, so we have formulated the sentence more carefully and delete the last sentence because of the length. Page 3 L65-71:

The rationale for this study is to assess the usability and utility of the proposed clinical framework as well as to identify possible differing somatosensory and psychosocial phenotypes between the subgroups. This could increase our knowledge of the underlying pain mechanisms of neck-arm pain. The longitudinal analysis of the subgrouped participants may help to assess possible predictors for pain persistency.

Reviewer 2 

Abbriviations are commonly not used in abstracts.

Thank you for the comments. The PLOS ONE author guidelines state that abstracts should not include abbreviations, if possible. We have revised the abstract and used only one abbreviation (QST) in order to reduce word count. We hope this is acceptable.

Reference no 1: Vos, T., et al., 2012 is based on data from 1990-2010. Maybe newer references are available? Same thing with reference no 3: Cote, P., et al., 2004 (also here newer date might be available. 

We have added more recent literature, and adapted the first sentence. Please see page 4, L73-74.

Reference:

Global, regional, and national incidence, prevalence, and years lived with disability for 354 diseases and injuries for 195 countries and territories, 1990-2017: a systematic analysis for the Global Burden of Disease Study 2017. Lancet, 2018. 392(10159): p. 1789-1858.

As reviewer 1 requested to shorten the introduction, we have deleted the sentence with the reference of Cote et al 2004. Page 4, L74-76.

page 7: the aim: Hence, the overall goal of the current study is the evaluation of a mechanism-based classification for patients with neck-arm pain.

This study has not completed the sampling, thus not being able to evaluate this classification. This study has more characteristic of a protocol. The editor should determine if the aim should be modified or if protocols without results are allowed. Furthermore, a modification of the title could be relevant, since an evaluation has not yet taken place in this study. 

Thank you. This submitted manuscript is a study protocol and describes a planned study. 

We changed the title from: 

Study Protocol: Evaluation of a mechanism-based classification for neck-arm pain – A cross sectional and longitudinal study

 into: 

Application and utility of a clinical framework for spinally referred neck-arm pain : A study protocol of a cross-sectional and longitudinal study

PLOS ONE does accept submission and publication of study protocols without results. For further info, please see the link below:

https://journals.plos.org/plosone/s/other-article-types#loc-registered-reports

Page 9: Sampling and Inclusion criteria: unilateral pain is only inclusion criteria, but what about how long time the pain has been present (acute, subacute, chronic)? This is vital information in order to replicate and understand why and how this classification will be developed. 

Thank you, this is an interesting comment. During the clinical examination, we will collect all relevant data, including the duration of symptoms . However, symptom duration is not relevant for the patient subgrouping, based on the clinical framework. The clinical framework should be applicable irrespective if patients have an acute, subacute or chronic pain condition. After data completion we will investigate if there are differences between and/or within the groups in terms of stage of disorder and if this had any bearing on their outcomes.

What about interventions that the included people may take up during the 12 months (fx physiotherapy, pain management, acupuncture etc.), how are these integrated/taken into account in the data?.

Thank you, this is an important point. On page 15, L373-374 we wrote that we ask in a follow-up questionnaire for “Pain behavior, disability and therapies carried out”. So we will capture the interventions (operations, physiotherapy, exercises, pain management, infusions etc.) to get an idea which potential factors that may have changed the pain over 12 months.

We changed the wording “therapies” to “interventions” and added the above mentioned examples in brackets. 

page 14: sample size: A sample size of 45 per group is 351 required to detect a medium sized effect......

Effect is not an aim of this study, maybe the authors could clarify why 45 subjects should attend each subgroup when intervention is not a part of this study, but rather the characteristics/symptoms of the patients/between patients. 

Thank you for the comment, we don’t exactly understand the number of 351 that you refer to. We haven´t used this number in our manuscript for the sample size calculation.

The term effect size as we use it here is not supposed to address effectiveness of an intervention, but as a measure of magnitude in differences between groups with respect to our main outcome. We have explained this further in the section: “Sample size”. P 16 L 379-384.

Figure 2: A bit confusing that one box/characteristic is half green/half orange, and another one third green and two-third orange (they are actually both mixed). Can this be explained below the figure?

And the grey box should also be explained like the others colors below the figure (the pain characteristics).

Thank you for your feedback. We have inserted further information on Page 11 L 270-272.

However, some conditions present with neuropathic pain features, e.g pain descriptors such as tingling, pins and needles, burning, pain attacks and numbness, hence the color coding to present mixed pain types.

We amended the legion of Figure 2. It reads now as follows on page 11/12 L274-279:

Fig. 2 Clinical framework of spinally referred neck-arm pain, adapted from reference. The color code represents characteristics of nociceptive or neuropathic pain. The green color represents nociceptive pain characteristics, the orange color represents neuropathic pain characteristics (e.g. paraesthesia, numbness, pain descriptors such as burning pain, pain attacks). Green and orange represent mixed pain characteristics. The dark grey color represents a radiculopathy, which is not defined by pain, but by the presence of a neurological deficit. 

Reviewer 3 

It is an interesting topic but it would be great to add more rationale for doing this study: 1) how it would change rehabilitation/physiotherapy practices?; 

Thank you very much for your comment!

We wrote in our introduction that studies have shown that tailored treatment is more effective than standard therapy in patients with low back pain [3-8]. Further, the study by Schaefer et al demonstrated that subgroup specific targeted treatment in patients with low back-related leg pain was superior for the specific subgroup compared to other subgroups.

The same may be applicable in patients with spinally referred neck-arm pain.

We have revised the introduction and hope that our rational given for the study is satisfactory.

Please see Page 4 L97-153.

2) What limitations are there for this study?

We added a paragraph on limitations of the study, please see on page 17 421-427:

A limitation of this study is that the patient allocation to subgroups is based on clinical examination findings. Additional instrumental measurements (e.g. MRI, nerve conduction studies, Somatosensory Evoked Potentials) to validate clinical findings will unlikely be available. Furthermore, the clinical examination and patient classification will be performed by one examiner. An assessment by a second examiner would enhance the validity of the study, however there are resource limitations, plus repeated assessment would impose a considerable burden to the patient.

Reviewer 4 

Review of Study Protocol: Evaluation of a mechanism-based classification for neck- arm pain. A cross sectional and longitudinal study

This is a well-conceived study protocol to evaluate a proposed classification system for neck-arm pain. I enjoyed reading it. 

Thank you very much for your encouraging feedback.

Does the manuscript provide a valid rationale for the proposed study, with clearly identified and justified research questions?

The paper provides a valid, and thorough rationale for the proposed study. The research questions are well formed and follow the logic of the background and introduction section.

In the abstract, the rationale is described by the single-sentence – “The rationale for this study is to assess the usability of the proposed classification system” – 

I would like to see this expanded to say that in addition to its usability the rationale is to assess its utility and robustness too by assessing the classification longitudinally in participants whose pain syndrome is expected to change over time. 

We added your recommendation in the abstract Page 3 L65-69:

The rationale for this study is to assess the usability and utility of the proposed clinical framework as well as to identify possible differing somatosensory and psychosocial phenotypes between the subgroups. This could increase our knowledge of the underlying pain mechanisms of neck-arm pain. The longitudinal analysis of the subgrouped participants may help to assess possible predictors for pain persistency.

In the introduction L112-113 I would like to have read why “The use of a mechanism-based classification should be applied to direct assessment and management [12].” In contrast to the other classification systems also described – could they not equally direct assessment and management? 

Thank you for this input. The other classifications mentioned, such as classification in stage of disorder, symptom duration, pain severity, location of pain or specific and nonspecific neck pain do not address the source of symptoms and the underlying pain mechanisms, which both are important to consider in the management of patients with neck-arm pain. We have revised the introduction and added this comment, please see page 6, L131-153.

The Schmid and Tampin classification approach is introduced as the preferred classification model you are deciding to utilise and evaluate which is logical as you make the argument that a mechanism-based approach is missing – but is the Schmid and Tampin approach the only one – are there others? And if so, why this one? 

Thank you, this is a good point. We tried to explain this on page 6 L146-153:

This clinical framework for spinally referred pain will be used to build the subgroups for investigating the somatosensory and psychosocial profiles of neck-arm pain patients in our study. The advantage of this framework over other mechanisms based approaches [3, 9-14] is the precise terminology, the differentiation between radicular pain and radiculopathy and based on this a differentiated mixed pain presentation.

Paragraph (starting on line 123) introducing QST – I think you could be much more explicit here in arguing that including this approach will allow some resolution between mixed types of pain experience. If this is correct, I’d say make this assertion as a goal of the study nearer the beginning of the paragraph – then go on to explain why. Similarly, in the paragraph preceding this one – the aim is mentioned at the end of the paragraph, and I think the manuscript would benefit from these statements being introduced earlier and therefore in a bolder way. Overall the rationale is good, but I think it could be crafted to tell an even more compelling story with some minor edits and removal of passive voice – I have alluded to specific edits/typos below. 

Thank you. We followed your recommendation and revised the manuscript on page 5, L 097-116.

We introduced the paragraph on QST earlier in the paper.

Is the protocol technically sound and planned in a manner that will lead to a meaningful outcome and allow testing the stated hypotheses?

The manuscript should describe the methods in sufficient detail to prevent undisclosed flexibility in the experimental procedure or analysis pipeline, including sufficient outcome-neutral conditions (e.g. necessary controls, absence of floor or ceiling effects) to test the proposed hypotheses and a statistical power analysis where applicable. 

As there may be aspects of the methodology and analysis which can only be refined once the work is undertaken, authors should outline potential assumptions and explicitly describe what aspects of the proposed analyses, if any, are exploratory.

Thank you for the comment. It is true, in certain points we have tried to stay flexible. But only in points which do not influence the investigation and the examination procedure. We have tried to describe this part more explicitly. See Page 17 L392-418.

The ethics and design sections are well written and clear. I think in Figure 1 the vertical title bar “Longitudinal Sectional” should just read “Longitudinal”. The protocol is technically sound, and the measurement and analytical approaches are justified and appropriate.

Thank you, we corrected this.

The sample size calculation sub-section is welcome (L349)– however it lacks detail in that it does not stipulate the target effect size the calculation is based on (the parameter, the type, or the magnitude), nor how the effect size was was determined. 

Thank you. Reviewer 1 also made this comment (see above) and we added further information on Page 16 L379-384.

Reference to G-power should include detail of the software and the version number used for completeness.

A minor style point, but I think the software used should be detailed at the end of the paragraph, not as the first thing (likewise in the “statistical analysis” sub-section). 

Thank you, we made amendments accordingly see page 16 L379.

Calculation of the sample size was processed with G-Power (Version: G*Power 3.1.9.4.).

Is the methodology feasible and described in sufficient detail to allow the work to be replicable?

The methods are feasible. I note that sampling is due to have started in June 2020, and also note that no allusion to the possible effects prolonged social restrictions due to the COVID pandemic might have on recruitment, retention and practicality of measuring constructs that cannot be done with participants remotely – i.e. QST assessment - in Germany. If there has been any mitigation for these unforeseen circumstances, then I think they should be added to the protocol. 

Thank you, this is a good point. We added the following sentence. Page 9 L220-221.

During the whole test procedure the hygiene guidelines of the Robert Koch Institute and the University of Applied Sciences Osnabrueck are followed due to the COVID pandemic.

And L218-219:

The recruitment is anticipated to take place is planned from July 2020 until probably December 2021, depending on the COVID pandemic.

I think L216-218 “… as well as the information sheet, the informed consent and a link to the study's homepage (www.nacken-armschmerzen.de)” might be better as “… as well as familiarisation information including a written information sheet, the informed consent material and a link to the study's homepage …”

Informed consent information is clearly stated in the Ethics sub-section, so the repeated allusion to it in L219-20 feels unnecessary. 

Thank you, we have included your recommendation , please see at page 10 L233-237.

Testing Protocol subsection – the first sentence is lacking a point - when or where will this take place, will the listed tasks be undertaken in one appointment for example? The next sentences provide more detail, but I think this sub-section could be clearer.

L227 – suggest signposting the reader here that clinical self-reported questionnaire details are provided below

We had documented under “Setting” that the measurements will take place at the INAP/O at the University of Applied Science Osnabrueck. 

We have added further information in the testing protocol.

See Page 9 L 217-218.

I note there is no allusion in the inclusion criteria for age; presumably the study is focused on adults and therefore needs at least a minimum age.

Thank you, that is correct. We have added the information on page 9/10 L226-227.

The new sentence is:

The inclusion criterium is unilateral neck-arm pain in participants aged between 18 – 75 years.

The exclusion criteria are comprehensive but appear to be an exclusion based on any count of pre-determined systemic pathologies (in contrast to say a comorbidity index threshold, or health-care utilisation threshold); I am not familiar with the specific pain syndrome under investigation in this paper so I do not know if there is room to justify this approach or the selection of the pathologies with due referencing, this would also defend the approach from the point of view of over-excluding and therefore only recruiting and testing a niche sub-set of participants– so my question would be are these exclusions typical in the field?. 

One particular query I have is “… elbow or hand disorders in the last months, …” – is this referring to musculoskeletal disorders or other disorders (neurological, or peripheral vascular for example), and a definitive number of months would make the exclusion clearer.

Thank you. The exclusion criteria are comprehensive, however they are needed in order to guarantee that (i) participants have spinally referred neck-arm pain and not pain of other origin; (ii) QST measurements are not influenced by other factors. For example, a thyroid dysfunction or diabetes may cause sensory alterations.

We want to make sure that potential measurements on the arm are not falsified by additional localized pain at the shoulder, elbow or hand from conditions like fractures, epicondylopathia etc.. Therefore we have excluded these.

The Classification system sub-section is welcome – a minor point is that while neural mechanosensitivity and neurodynamic tests are mentioned in the text prior to the bullet-points in this sub-section, the neurological integrity test is not and might confuse readers unfamiliar with the classification system – it might help to weave this into the narrative prior to the bullets at least in the clinical examination sub-section. 

We had documented the neurological examination in the clinical examination sub-section. We have specified this and added the term neurological integrity. Page 11, L258-259. 

Clinical Examination sub-section – I would like to see more detail referred to either in the text or as a footnote (if the journal allows it) or as supplementary material. 

Specifically; what are the red-flag questions? 

what are the anchors of the sleeping numerical rating scale? 

The passive and active c-spine/shoulder complex exam – while these are referenced, I would like to see in the text what movements/planes are planned to be measured. 

We have added the red flag information and the information about the sleeping NRS directly in the manuscript (Page 10/11 L251-256). Further information about the clinical examination we will add as supporting information (S1 File).

L296 “Kappa values were between 0.70 and 0.96 [70]” would be a stronger statement with a justification and an interpretation of these data 

Thank you, we added the interpretation to each Kappa, ICC and Cronbach’s α. Page 12/13 L300-321.

Statistical Analysis sub-section - L360-361 – “In case of violation of statistical assumptions data transformation or non-parametric testing will be considered” please provide details of what transformations will be considered or soften with “appropriate” transformations. L374-376 – this sentence is welcome but needs to be re-crafted, so it is clearer.

We added the missing information Page 16 L396-399:

In case of violation of statistical assumptions data appropriate transformation (e.g. log-Transformation) or non-parametric testing will be considered such as Kruskal-Wallis-Test, Friedman Test and non-parametric or robust regression.

The Conclusion I think is excellent. 

Thank you.

Have the authors described where all data underlying the findings will be made available when the study is complete?

Yes 

Is the manuscript presented in an intelligible fashion and written in standard English?

Yes. I did find some trivial edits which are outlined below:

Abstract, L61 – is there a need to pluralise regression model to models here? 

Thank you, we have corrected this. 

Multiple regression models will be used to analyze potential predictors for the clinical course.

Introduction, L97 - “Studies showed that tailored treatment was more effective …” might be better as “Studies have shown that tailored treatment is more effective …” 

Studies have shown that tailored treatment is more effective than standard therapy in patients with low back pain [21-26].

Introduction, L103 – “condition” should be pluralised I think 

Similarly, there is support for a mechanism-based management approach for some neuropathic pain conditions

Introduction, L128 –suggest change “were” to “have been” 

Based on QST, differences in somatosensory profiles between patients with C6/7 cervical radiculopathy and patients with C6/7 radicular pain without radiculopathy have been documented [15, 16].

Introduction L140-145 – while understandable, this is a long sentence to parse – it might help a reader to break into >1 sentence 

Various clinical predictors of pain persistency in patients with neck and neck-arm pain have been reported [17-25]. This includes psychological and cognitive-behavioural factors such as post-traumatic stress and pain catastrophizing in patients with whiplash and subacute neck pain [21], an initial high level of self-reported pain and disability [18, 22], older age and a history of other musculoskeletal disorders in nonspecific neck-arm pain (NSNAP) [26] [18, 22].

Introduction L145 – I think there is a missing “of” before “… a new episode …”

Introduction L145 – I think “… and …” could be changed to “… as was …” to help this sentence. 

Poor muscle endurance as was depressed mood were factors for development of a new episode of pain in nonspecific chronic neck pain [19].

Introduction L149 – I think the sentence starting “One single …” could be made clearer and link to the next sentence better; in fact I think “To date, there is only this one study that …” could be bolder and say “To date, this is the only study that …” in the next sentence 

One single study, assessing patients with chronic neck and neck-arm pain with QST (CPT, PPT), clinical tests (neurodynamic tests), psychosocial factors (PCS, DASS-21), functional questionnaires (NDI) and neuropathic screening tools (SLANSS), demonstrated that baseline neck disability, comorbidities and higher psychological distress contributed to predicting disability at 12 months [17]. To date, this is the only study that has collected quantitative sensory and clinical tests to investigate potential predictors of chronic neck pain [17].

Introduction L155 – This final sentence could also be made bolder, perhaps choose another way of saying “to date” in it, and not end with a passive goal of the proposed study. 

Thank you! 

However, no study to date has included clinical measurements (e.g. active and passive cervical movement impairments, neurodynamic tests) as well as the somatosensory and psychosocial profile based on classified subgroups of different underlying pain disorders, to investigate potential predictors, as proposed in this current study.

Introduction L161 – I would consider changing the rather passive opening to this paragraph (“It could be summarized that …”) with something more assertive. 

Thank you, we delete the paragraph starting with: (“It could be summarized that …”) also to make the introduction shorter. Now the final sentence before specific aims is:

Hence, the overall goal of the current study is the evaluation of the clinical framework for referred pain in patients with neck-arm pain and their somatosensory and psychosocial profile as well as the clinical course over time.

Page 8 L181-183.

Methods L200 – I am not familiar with “executive sample”, is this correct? Should it be a sample of convenience, or a volunteer sample 

We changed the sentence like this: The study population will be recruited as an executive sample, Each subject who fulfils the eligibility criteria will be included.

Methods L208 – I think “criterium” should be “criterion”

Methods L216 – tense consistency; I think “… is …” should be “… will be …” 

Thank you, we correct it: The inclusion criterion

Following the screening, each suitable patient will be given an appointment for the clinical examination…

Methods L220 – I think “..., in another parallel …” could be simply We shortened the sentence and hope that goes in the right direction:

Healthy controls participants for the neck-arm area will be recruited in another parallel study. conducted at the University of Applied Sciences Osnabrueck, which will start at the same time as the cross-sectional study. 

Methods L268 – “Since the ‚radiculopathy group’ …” should be “Since the ‘radiculopathy group’ …” 

Thank you! Since the ‘radiculopathy group’

Methods L277 (& L280; L287) – suggest to remove the second , the use of the word “good” for the value of 0.73 needs would be strengthened with a published precedent , I would also like to see the type of ICC referred to in this line for completeness (also for ICC on L285) 

Changed in: 

The internal consistency showed good values with acceptable Cronbach's α of α = 0.73 and a good ICC (2,1) of 0.81. A median of 40 can be considered a severe kinesiophobia [29].

And:

ICC (2.1) of 0.81 [30].

Methods L279 – “The Depressions- Anxiety- Stress- Scale (DASS) …”, I think this should read “The depression anxiety and stress scale (DASS) …” And, “… examines 21 Items …” should be “… examines 21 items …” Thank you, we changed it:

The depressions- anxiety- stress- scale (DASS) [31] examines 21 items,

Methods L282 – I do not think the word “the” is needed in “… monitors intensity of the pain …” 

The Neck Disability Index (NDI) monitors intensity of the pain,

Methods L293 – “The pretest was …” could be better stated as “A pretest was …”

L300: A pretest was successfully completed and…

Methods L304-305 the statement “… by a second blinded examiner toward the classified subgroup, …” would be clearer as “… by a second examiner who will be blinded to the subgroup classification process , …” or similar 

Thank you, that sound much better!

… by a second blinded examiner who will be blinded toward the subgroup classification process, the

Methods L306 - I think “… seven tests which tests 13 different …” should be “… seven tests that assess 13 different …” …includes seven tests that assess 13 different somatosensory

Methods L307 – I think “… Baseline temperature is at …” could be better as “… Baseline temperature will be set at …” Baseline temperature will be set at 32°, cutoff temperatures are 5°C and 50°C.

Methods L312 – I think “… temperature of 3 measurements will be calculated.” could be “… temperature from 3 measurements will be calculated.”

 … threshold temperature from 3 measurements…

Methods L321 – “Subjects are asked …” should be “Subjects will be asked …”

Subjects will be asked….

Methods L355-358 – This sentence does not make complete sense to me 

We changed the sentence and hope it is more clear now page 16 L387-390:

All data analysis will be performed with SPSS and R. Characteristics of study population depending on subgroups will be done by descriptive statistics, ANOVA, Kruskal-Wallis-Test or Chi²Test depending on the scaling of variables and statistical assumptions of parametric testing.

Methods L358 – the word “answered” needs to be changed

 … Objective (I,II) will be addressed by…

Methods L360-361 – “In case of violation of statistical assumptions data transformation or non-parametric testing will be considered” please provide details of what transformations will be considered, or soften with “appropriate” transformations We added the missing information:

…(M)AN(C)OVA and/or (hierarchical) regression models will be used. In case of violation of statistical assumptions data transformation (e.g. log-Transformation) or non-parametric testing will be considered such as Kruskal-Wallis-Test, Friedman Test and non-parametric or robust regression.

Methods L368 – “… ANOVA of repeated measurements …” should be “… repeated-measures ANOVA …” I think. MANOVA, repeated-measures ANOVA,

This is a valid protocol for a needed study and utilises both a cross sectional and longitudinal design. The protocol is sound, and my comments really are of style and some content detail which is missing in my opinion. With some minor revision, the manuscript protocol would be a welcome addition to the literature. 

Thank you very much!

Literature:

1. Ottiger-Boettger K., B.N., Landmann G., Stockinger L., Tampin B., Schmid A., Somatosensory Profiles in Patients with non-Specific Neck-Arm Pain with and without positive Neurodynamic Tests. Musculoskeletal Science and Practice, 2020. In Press.

2. Rolke, R., et al., Quantitative sensory testing in the German Research Network on Neuropathic Pain (DFNS): standardized protocol and reference values. Pain, 2006. 123(3): p. 231-43.

3. Moloney, N.A., et al., The clinical utility of pain classification in non-specific arm pain. Man Ther, 2015. 20(1): p. 157-65.

4. Costa Lda, C., et al., Primary care research priorities in low back pain: an update. Spine (Phila Pa 1976), 2013. 38(2): p. 148-56.

5. Vibe Fersum, K., et al., Efficacy of classification-based cognitive functional therapy in patients with non-specific chronic low back pain: a randomized controlled trial. Eur J Pain, 2013. 17(6): p. 916-28.

6. Kent, P., H.L. Mjosund, and D.H. Petersen, Does targeting manual therapy and/or exercise improve patient outcomes in nonspecific low back pain? A systematic review. BMC Med, 2010. 8: p. 22.

7. O'Sullivan, P., Diagnosis and classification of chronic low back pain disorders: maladaptive movement and motor control impairments as underlying mechanism. Man Ther, 2005. 10(4): p. 242-55.

8. Schafer, A., et al., Outcomes differ between subgroups of patients with low back and leg pain following neural manual therapy: a prospective cohort study. Eur Spine J, 2011. 20(3): p. 482-90.

9. Dewitte, V., et al., Subjective and clinical assessment criteria suggestive for five clinical patterns discernible in nonspecific neck pain patients. A Delphi-survey of clinical experts. Man Ther, 2016. 26: p. 87-96.

10. Rasmussen, H., et al., In a secondary care setting, differences between neck pain subgroups classified using the Quebec task force classification system were typically small - a longitudinal study. BMC Musculoskelet Disord, 2015. 16: p. 150.

11. Childs, J.D., et al., Neck pain: Clinical practice guidelines linked to the International Classification of Functioning, Disability, and Health from the Orthopedic Section of the American Physical Therapy Association. J Orthop Sports Phys Ther, 2008. 38(9): p. A1-a34.

12. Liu, R., et al., Classification and Treatment of Chronic Neck Pain: A Longitudinal Cohort Study. Reg Anesth Pain Med, 2017. 42(1): p. 52-61.

13. Werneke, M., D.L. Hart, and D. Cook, A descriptive study of the centralization phenomenon. A prospective analysis. Spine (Phila Pa 1976), 1999. 24(7): p. 676-83.

14. Yarznbowicz, R., M. Wlodarski, and J. Dolutan, Classification by pain pattern for patients with cervical spine radiculopathy. J Man Manip Ther, 2019: p. 1-10.

15. Tampin, B., H. Slater, and N.K. Briffa, Neuropathic pain components are common in patients with painful cervical radiculopathy, but not in patients with nonspecific neck-arm pain. Clin J Pain, 2013. 29(10): p. 846-56.

16. Tampin, B., et al., Quantitative sensory testing somatosensory profiles in patients with cervical radiculopathy are distinct from those in patients with nonspecific neck-arm pain. Pain, 2012. 153(12): p. 2403-14.

17. Moloney, N., et al., Are measures of pain sensitivity associated with pain and disability at 12-month follow up in chronic neck pain? Musculoskeletal Care, 2018.

18. Carroll, L.J., et al., Course and Prognostic Factors for Neck Pain in Whiplash-Associated Disorders (WAD). European Spine Journal, 2008. 17(S1): p. 83-92.

19. Shahidi, B., D. Curran-Everett, and K.S. Maluf, Psychosocial, Physical, and Neurophysiological Risk Factors for Chronic Neck Pain: A Prospective Inception Cohort Study. J Pain, 2015. 16(12): p. 1288-1299.

20. Walton, D.M., et al., Risk factors for persistent problems following acute whiplash injury: update of a systematic review and meta-analysis. J Orthop Sports Phys Ther, 2013. 43(2): p. 31-43.

21. Sterling, M., et al., Prognosis after whiplash injury: where to from here? Discussion paper 4. Spine (Phila Pa 1976), 2011. 36(25 Suppl): p. S330-4.

22. Walton, D.M., et al., An Overview of Systematic Reviews on Prognostic Factors in Neck Pain: Results from the International Collaboration on Neck Pain (ICON) Project. Open Orthop J, 2013. 7: p. 494-505.

23. Park, S.J., et al., Factors associated with increased risk for pain catastrophizing in patients with chronic neck pain: A retrospective cross-sectional study. Medicine (Baltimore), 2016. 95(37): p. e4698.

24. Pedler, A. and M. Sterling, Patients with chronic whiplash can be subgrouped on the basis of symptoms of sensory hypersensitivity and posttraumatic stress. Pain, 2013. 154(9): p. 1640-8.

25. Treede, R.D., The role of quantitative sensory testing in the prediction of chronic pain. Pain, 2019. 160 Suppl 1: p. S66-s69.

26. Rampakakis, E., et al., Real-life assessment of the validity of patient global impression of change in fibromyalgia. RMD Open, 2015. 1(1): p. e000146.

27. Rusu, A.C., et al., Fear of movement/(Re)injury in low back pain: confirmatory validation of a German version of the Tampa Scale for Kinesiophobia. BMC Musculoskelet Disord, 2014. 15: p. 280.

28. Woby, S.R., et al., Psychometric properties of the TSK-11: a shortened version of the Tampa Scale for Kinesiophobia. Pain, 2005. 117(1-2): p. 137-44.

29. Lundberg, M., J. Styf, and B. Jansson, On what patients does the Tampa Scale for Kinesiophobia fit? Physiother Theory Pract, 2009. 25(7): p. 495-506.

30. Cramer, H., et al., Validation of the German version of the Neck Disability Index (NDI). BMC Musculoskelet Disord, 2014. 15: p. 91.

31. Nilges, P. and C. Essau, [Depression, anxiety and stress scales: DASS--A screening procedure not only for pain patients]. Schmerz, 2015. 29(6): p. 649-57.

---

## [Decision Letter · Decision Letter 1]

6 Nov 2020

PONE-D-20-20814R1

Application and utility of a clinical framework for spinally referred neck-arm pain: A study protocol of a cross-sectional and longitudinal study

PLOS ONE

Dear Dr. Kapitza,

Thank you for submitting your manuscript to PLOS ONE. After careful consideration, we feel that it has merit but does not fully meet PLOS ONE’s publication criteria as it currently stands. Therefore, we invite you to submit a revised version of the manuscript that addresses the points raised during the review process.

We look forward to receiving your revised manuscript.

Kind regards,

Alison Rushton

Academic Editor

PLOS ONE

Additional Editor Comments (if provided):

Please address these few further minor points from the reviewers.

1. The title - As it stands, the changed title might benefit from some simplification to: "Application and utility of a clinical framework for spinally referred neck-arm pain: A cross-sectional and longitudinal study protocol"

2. In addition, the word "protocol" does not feature in the abstract which might be misleading

3. Ethics - first sentence would be improved by stating "The study has received ethical approval form the … "

4. Quantitative Sensory Testing (QST) section (L327 of tracked changes version), there is a "the" missing before "German ..."

5. L364 of tracked changes version, Lime survey - is propitiatory software/a company that, if so, would benefit from having company details/version nos included.

6. L387 of tracked changes version, adding company details/version nos to the SPSS and R allusion would be more complete(less...)

Reviewers' comments:

Reviewer's Responses to Questions

**Comments to the Author**

1. Does the manuscript provide a valid rationale for the proposed study, with clearly identified and justified research questions?

Reviewer #3: Yes

Reviewer #4: Yes

2. Is the protocol technically sound and planned in a manner that will lead to a meaningful outcome and allow testing the stated hypotheses?

Reviewer #3: Yes

Reviewer #4: Yes

3. Is the methodology feasible and described in sufficient detail to allow the work to be replicable?

Reviewer #3: Yes

Reviewer #4: Yes

4. Have the authors described where all data underlying the findings will be made available when the study is complete?

Reviewer #3: Yes

Reviewer #4: Yes

5. Is the manuscript presented in an intelligible fashion and written in standard English?

Reviewer #3: Yes

Reviewer #4: Yes

6. Review Comments to the Author

You may also provide optional suggestions and comments to authors that they might find helpful in planning their study.

Reviewer #3: Authors have provided satisfactory answers to reviewers' comments. The revised manuscript provides more clarification about the study protocol.

Reviewer #4: Thank you for your revised manuscript. I think the paper is improved and now reads well as a thorough protocol rationale and description based on your comprehensive responses to mine, and my fellow reviewers' thoughts. I have nothing more to add apart from from minor typos/edits which I have passed to the editor.

7. PLOS authors have the option to publish the peer review history of their article (what does this mean?). If published, this will include your full peer review and any attached files.

Reviewer #3: No

Reviewer #4: **Yes: **Gareth D. Jones

---

## [Author Response · Author response to Decision Letter 1]

1 Dec 2020

1. The title - As it stands, the changed title might benefit from some simplification to: "Application and utility of a clinical framework for spinally referred neck-arm pain: A cross-sectional and longitudinal study protocol"

Thank you for your comment.

We changed the title to: Application and utility of a clinical framework for spinally referred neck-arm pain: A cross-sectional and longitudinal study protocol

2. In addition, the word "protocol" does not feature in the abstract which might be misleading

Thank you very much. We added the following sentence in the Method section of the abstract part L 35:

We describe a study protocol.

3. Ethics - first sentence would be improved by stating "The study has received ethical approval form the … "

We added your recommendation, Page 7 L 160-161: The study has received ethical approval from the was approved by the Ethics Committee of the University of Applied Sciences Osnabrück (HSOS/2019/2/2)

4. Quantitative Sensory Testing (QST) section (L327 of tracked changes version), there is a "the" missing before "German ..."

Thank you. We added the missing word page 12 L289-290:

Standardized QST will be performed according to the reliable QST protocol of the German Research Network on Neuropathic Pain (DFNS)

5. L364 of tracked changes version, Lime survey - is propitiatory software/a company that, if so, would benefit from having company details/version nos included.

We added the missing information on page 13 L 326:

The three, six and 12 months follow-up will be conducted via postal hard copy and lime survey (Version: 3.22.210 + 200804).

6. L387 of tracked changes version, adding company details/version nos to the SPSS and R allusion would be more complete(less...) 

We added the missing informations page 14 L349:

All data analysis will be performed with SPSS (Version: 26) and R (Version: 3.6.3.).

---

## [Editor Report · Decision Letter 2]

4 Dec 2020

Application and utility of a clinical framework for spinally referred neck-arm pain: A cross-sectional and longitudinal study protocol

PONE-D-20-20814R2

Dear Dr. Kapitza,

We’re pleased to inform you that your manuscript has been judged scientifically suitable for publication and will be formally accepted for publication once it meets all outstanding technical requirements.

Kind regards,

Alison Rushton

Academic Editor

PLOS ONE

Additional Editor Comments (optional):

Thank you for addressing all final comments from reviewers.

---

## [Editor Report · Acceptance letter]

10 Dec 2020

PONE-D-20-20814R2 

Application and utility of a clinical framework for spinally referred neck-arm pain: A cross-sectional and longitudinal study protocol 

Dear Dr. Kapitza:

I'm pleased to inform you that your manuscript has been deemed suitable for publication in PLOS ONE. Congratulations! Your manuscript is now with our production department. 

Kind regards, 

on behalf of

Professor Alison Rushton 

Academic Editor

PLOS ONE